# "Running with cancer": A qualitative study to evaluate barriers and motivations in running for female oncological patients

Alice Avancini[1]*, Kristina Skroce[2], Daniela Tregnago[3], Paolo Frada[2], Ilaria Trestini[3], Maria Cecilia Cercato[4], Clelia Bonaiuto[3], Cantor Tarperi[2,5], Federico Schena[2], Michele Milella[3], Sara Pilotto[3], Massimo Lanza[2]

1 Department of Medicine, Biomedical, Clinical and Experimental Sciences, University of Verona Hospital Trust, Verona, Italy, 2 Department of Neurosciences, Biomedicine and Movement Sciences, University of Verona, Verona, Italy, 3 Department of Oncology, University of Verona Hospital Trust, Verona, Italy, 4 Epidemiology and Cancer Registry Unit, Regina Elena National Cancer Institute, IRCCS, Rome, Italy, 5 Department of Clinical and Biological Sciences, University of Turin, Turin, Italy

* alice.avancini@univr.it

**Data Availability Statement:** The data contained in the paper constitute our minimal underlying data set.

## Abstract

Nowadays, it is widely acknowledged that low physical activity levels are associated with an increase in terms of both disease recurrence and mortality in cancer survivors. In this light, deciphering those factors able to hamper or facilitate an active lifestyle is crucial in order to increase patients' adherence to physical activity. The purpose of this study was to explore barriers and motivations in a sample of female oncological patients, practising running using the ecological model and compare them with healthy controls. Focus group interviews were conducted at Verona University. Participants were 12 female cancer survivors and 7 matched healthy controls who had participated at "Run for Science" project. The interviews were transcribed verbatim and analyzed using content analysis. Transcripts were categorized according to the ecological model, identifying barriers and motivations as themes. About motivations, three sub-themes were included: *personal, interpersonal and environmental/organizational factors*. Regarding barriers, another sub-theme was recognized: *community/policy factors*. Compared to healthy controls, survivors expressed motivations and barriers specifically related to their oncological disease. Running was a challenge with their cancer and a hope to give to other patients. Main barriers were represented by treatment-related side effects, inexperienced trainers and external factors, e.g. delivery of incorrect information. Running programs dedicated to oncological patients should consider intrinsic obstacles, related to cancer and its treatment. The interventions should offer a personalized program performed by qualified trainers, together with a motivational approach able to improve participants' adherence to an active lifestyle.

## Introduction

In Italy, one out of three women will experience an oncological disease during lifetime [1]. Cancer is the second most common chronic disease in female population and in 2018 more

**Funding:** The authors received no specific funding for this work.

**Competing interests:** The authors have declared that no competing interests exist.

than 1,870,000 women in Italy were living with a cancer diagnosis [1]. The introduction in clinical practice of innovative treatments have allowed cancer survivors to achieve an improved prognosis and quality of life. Nevertheless, cancer patients often experience important treatment-related side effects, involving both the physical and psychological spheres, having a potential prolonged impact on patients' condition even after therapy conclusion [2].

An increasing amount of studies has demonstrated that physical activity (PA) and exercise (EX) are safe and feasible in the oncological setting. PA can support standard therapies, helping cancer survivors in reducing their risk of recurrence and mortality [3]. PA and EX can facilitate the management of some disease- and treatment-related effects, as fatigue, nausea and vomiting, increasing patients' quality of life [4, 5]. Moreover, the EX and PA benefits include improvement in cardiorespiratory fitness, strength, flexibility and body composition [6, 7]. The American College of Sport Medicine recommends patients with cancer to avoid inactivity and engage in at least 90 min/week of moderate-intensity aerobic PA, with strength EX two times per week [2].

One of the most common type of aerobic PA is running, not only for its physical and physiological benefits, but also for its accessibility and simplicity. A recent report indicated that there were 17.1 million running participants during the 2015 running season [8]. Running is the most widespread PA also in the cancer setting with an acknowledged beneficial impact [8]. Running confers numerous cardiovascular, metabolic, musculoskeletal and neuropsychiatric benefits and is strongly associated with lower body weight and smaller waist circumference [8]. This PA is shown to increase life-longevity and is often recommended as prevention and control for various chronic diseases, including cancer. Previous studies have identified different factors related to running motivation, as the desire to affiliate with other runners, an increase in self-esteem, physical motives for general health benefits, improving quality of life, coping with negative emotions and many more [9]. Despite many positive aspects connected with a more active lifestyle, there are many barriers that can interfere with EX adherence, particularly speaking about running, which may be more physically and psychologically difficult than some other activities [10].

These motivations and barriers are connected not only with the momentary health status, but also with the previous health-related experiences [11]. Furthermore, individual behaviour may be influenced by many elements that interact with the person [12] [13]. This approach, also called *ecological model* assumes that individual competencies, intrapersonal relations, organisational or community structures and political choices can influence or determine the individual's behaviour [12] in many fields, including physical activity and lifestyle. To date, no study investigated barriers and motivations in female cancer survivors performing running and compared them with their healthy controls. Therefore, the aim of this study was to qualitatively investigate barriers and motivations, according to the ecological model, in a sample of female cancer survivors practising running and compare them with healthy controls.

## Materials and methods

### Design

We conducted a series of focus group sessions among female adults affected or not by cancer to qualitatively assess barriers and motivations towards running.

The study was approved by the local Ethical Committee (Department of Neurological, Neuropsychological, Morphological and Movement Science, University of Verona, Prot. No. 165038) and followed to Standards for Reporting Qualitative Research (SRQR) guidelines for qualitative research [14, 15].

## Participants and recruitment

A purposive sample was employed to recruit women who had participated at "Run for Science" (R4S) project [16]. Inclusion criteria for the oncological group (OG) were: female participant, had been diagnosed with cancer, being $\geq$ 18 years of age and participating in R4S event. Regarding healthy controls (HC), women participating at R4S, with absence of chronic disease and 18 years of age or older were considered eligible. The inclusion criteria were assessed by AA through the database of R4S.

Eligible women were contacted individually via email by the research team to introduce them the study. If they agreed to participate, AA contacted them by telephone to organize the interview. Written informed consent was obtained from included participants the day of the interviews, before starting the focus group. To protect participants' identity pseudonyms were used to report the data.

## The "Run for Science" project

The R4S, previously described [17], is a research project endorsed by the University of Verona, which involves Italian, European and American scientific institutions. The purpose of this event, coordinated by FS, CT, and KS, is to investigate several aspects regarding the effects of endurance running, and usually involves more than 200 volunteer runners every year.

## Data collection

Focus groups were held, from April 2019-July 2019, in a meeting room at Department of Neuroscience, Biomedicine and Movement of Verona University and lasted approximately 60 minutes. Overall, five focus groups were organized, three for oncological subjects (n = 4, 5 and 3) and two for healthy participants (n = 4 and 3). Interviews were conducted separately for the groups of women with a cancer diagnosis and the groups of healthy subjects. The reason for this choice was to make a more possible comfortable environment to bring out detailed information regarding own personal history.

The interviews were carried out by ML and observed by AA and PF. ML is Associate Professor in Sport Science and Methodology at Verona University with expertise in PA and health promotion. AA is a PhD student involved in EX in oncological patients, with previous interview experience and PF is a master's degree student in preventive and adapted PA. Participants were asked about barriers and motivators to running, applying the ecological model. AA and the ML developed some semi-structured questions, based on previous studies [18, 19] to guide the interviews (Table 1). The interview guide was reviewed by DT, the dedicated psycho-oncologist working at Oncology Department of Verona University Hospital. All interviews were audio-recorded and transcribed verbatim. Data collection continued until saturation principle was reached, i.e. no new information seemed to emerge from the interviews.

After each focus group session, a questionnaire to investigate the socio-demographic data (e.g. birth date, education level, marital status and occupational status) and clinical information (medical history) was provided to participants to complete. Perceived economic insecurity was assessed with the closed-ended question *"How do you get to the end of the month, with your available financial income?"* with four possible response (i.e. many difficulties/ some difficulties/ easily/ very easily).

## Analysis

ML, AA and PF independently analysed the data, using the content analysis. This approach was performed with Atlas.ti^TM software and involved a process of reading, reflection, decoding

**Table 1. Semi-structured interview questions.**

| *Motivations* |
| --- |
| • From the personal point of view (thinking of physical and psychological state and previous experience) is there any factor that in your opinion may motivate the adherence to running program? |
| • From the social point of view (thinking of relationships with other people, friends, colleagues, family) is there any factor that in your opinion may motivate the adherence to running program? |
| • From the environmental point of view (thinking of place, organizations and institutions) is there any factor that in your opinion may motivate the adherence to running program? |
| • From the cultural point of view (thinking of politics and national/regional rules) is there any factor that in your opinion may motivate the adherence to running program? |
| *Barriers* |
| • From the personal point of view (thinking of physical and psychological state and previous experience) is there any factor that in your opinion may limit the adherence to running program? |
| • From the social point of view (thinking of relationships with other people, friends, colleagues, family) is there any factor that in your opinion may limit the adherence to running program? |
| • From the environmental point of view (thinking of place, organizations and institutions) is there any factor that in your opinion may limit the adherence to running program? |
| • From the cultural point of view (thinking of politics and national/regional rules) is there any factor that in your opinion may limit the adherence to running program? |

and re-reading on the meaning of the data collected, in order to analytically interpret the text. First, the text was read several times to identify recurring ideas and to get a sense of the whole discussion. The second point included the formulation of codes summarizing the salient features of collected data. The third, was grouping the code into themes and eventually sub-themes. The final step involved all three authors with a process called *triangulation*. This consisted in presenting the emerged findings to the research team members, comparing the results and defining the final themes [20]. Moreover, the researchers compared the emerged themes from the HC and OG to find similarities and differences.

## Results

All the invited cancer survivors (n = 12) participated to the study, while only 7 out of 13 healthy females completed the focus group. Table 2 illustrates the socio-demographic and medical characteristics of both groups. The transcripts were analyzed according to the ecological model and the following common themes were categorized to reflect the levels: 1) motivations and 2) barriers in running.

### Theme 1: Motivations

Features that have stimulated participant's will to be or become active in everyday life, even after the conclusion of oncological treatments, include three main sub-themes: individual, interpersonal and organizational factors (Table 3).

**Individual factors.** Different aspects connected with running were common in both groups, such as enjoyment, previous experience, as well as mental and physical benefits of exercising. Some women experienced a true well-being during their running workout, as reported by this woman: *"I like running, I like the emotion of moving with my own legs in the environment, and the fatigue I feel is pleasant because it means that by this kind of practice I am moving towards my goal."* (Giovanna, OG). Other women perceived their workouts as a time of their everyday life where they enjoy themselves, as reported by this woman: *"For me, it is enjoyment and passion. I started practicing sport while I was not young anymore and I literally fell in love with running."* (Lara, HC). All women reported that their previous EX experience

**Table 2. Participant' characteristics.**

| | Oncological group (n = 12) | Healthy group (n = 7) |
|---|---|---|
| Age[a], mean (SD) | 50.5 (5.9) | 47.5 (8.0) |
| Body mass index[b], mean (SD) | 21.9 (2.8) | 22.1 (0.8) |
| Education, N | | |
| Secondary | 1 | 0 |
| High school degree | 7 | 4 |
| Undergraduate degree | 3 | 2 |
| Postgraduate degree | 1 | 1 |
| Marital status, N | | |
| Unmarried | 4 | 3 |
| Married | 7 | 4 |
| Divorced | 1 | 0 |
| Employment, N | | |
| Part time employed | 8 | 3 |
| Full time employed | 4 | 4 |
| Family income[c], N | | |
| Many difficulties | 1 | 0 |
| Some difficulties | 4 | 1 |
| Easily | 4 | 5 |
| Very easily | 3 | 1 |
| METs—Physical activity, mean (SD) | 3069.9 (1536.5) | 2441.3 (1119.1) |
| Tumor site, N | | |
| Colorectal | 2 | - |
| Hematologic | 1 | - |
| Breast | 9 | - |
| Stage, N | | |
| Unknown | 5 | - |
| Early | 4 | - |
| Advanced | 3 | - |
| Metastatic | 0 | - |
| Months from diagnosis, mean (SD) | 57.6 (34.5) | - |
| Undergone surgery, N | 11 | - |
| Undergone chemotherapy, N | 9 | - |
| Undergone radiation therapy, N | 8 | - |
| Undergone hormone therapy, N | 8 | - |
| Undergone others treatment, N | 0 | - |
| Current treatment status, N | | |
| Incoming | 0 | - |
| Ongoing | 0 | - |
| Ended | 12 | - |

SD, standard deviation, N, number; Mets, metabolic equivalent of the task expressed in minutes per week

[a] Expressed in years

[b] Expressed in units of kg/m$^2$

[c] Perceived economic insecurity assessed by the question: *How do you get to the end of the month, with your available financial income?*

represented a positive motivator in building and maintaining their active lifestyle. Although the mental health benefits from exercise represented a common factor detected in both groups,

**Table 3. Motivation and barriers related to running EX identified by cancer survivors compared to healthy controls.**

| Ecological model (level) | Motivations | | Barriers | |
|---|---|---|---|---|
| | Cancer survivors | Healthy controls | Cancer survivors | Healthy controls |
| Personal factors | • Prior EX experiences | • Prior EX experiences | • Lack of time (in progress) | • Lack of time |
| | • Enjoyment | • Enjoyment | • Injury | • EX failure |
| | • Physical and mental benefits | • Physical and mental benefits | • Cancer-related treatment side effects | |
| | • Cancer-related challenge | • Positive EX results | | |
| | • Hope for other patients | • Ex easy budget | | |
| Interpersonal factors | • EX group support | • EX group support | • Trainer not qualified | • Lack of social support |
| | • Family support | • Family support | | |
| | • Friends support | | | |
| | • Physician support | | | |
| Environmental and organizational factors | • Natural environment | • Natural environment | • Poor personal security | • Poor personal security |
| | • Organized training | | • Untended environment | • Untended environment |
| | | | • Air pollution | |
| Community and policy factors | | | • Traditionalist culture | • Running is underestimated compared to |
| | | | • EX only for athletes and body image | • other sports |
| | | | • Incorrect information delivery | |

origins and consequences were different. In particular, healthy subjects applied these benefits to deal with work, family or personal stress, as reported by Laura (HC): *"If I'm tired and exhausted at the end of my working day, I usually go for a run and reach some kind of mental regeneration."* In contrast, oncological patients benefitted from running experience in terms of better facing the prescribed treatments, as reported by Elisa (OG): *"I suffered a lot from the psychological point of view after radiotherapy and chemotherapy, but now I am feeling much better and as far as I understand this is due to my running workouts."* Other factors, such as the performance results connected to running, the fact that it is a cheap and easy to perform activity, were identified as personal motivation by the healthy group. In the oncological group, a crucial motivation was specifically related to the disease. In this regard, all the participants confirm that running means for them a personal challenge after cancer: *"My main motivation is to show to myself that I can do it, I can do something incredible, like a half marathon, even after my cancer."* (Nicoletta, OG). Another important aspect recognized as a potent stimulus to running is to give hope to other patients: *"I run to give hope to who is beginning the tumor winding path. Maybe they will see me and say: okay if she won it, I can do it too."* (Stefania, OG).

**Interpersonal factors.** The relationship with others was an important motivator highlighted during the focus group interviews, in both the oncological and healthy groups. Training with other people was recognized as a vehicle of sociality able to increase motivation in running. Moreover, for OG, exercising with someone who shares similar disease-related experiences, helped them to remain motivated and active: *"With these women I immediately found myself very well. We speak the same language because we share the same cancer history."* (Stefania, OG) and *"Even if I cannot go, I say to myself: no, someone is waiting for me, I cannot skip, I need to go and workout with them."* (Elisa, OG). Family support is common in both groups. In the HC perspectives, partner stimulate the participants to train, as Lara (HC) told: *"My husband encouraged me to run. He is a crucial support for me.".* In cancer survivors' group, the family support resulted overall positive, but sometimes controversial. Some of them were encouraged, as Margherita (OG) remembered: *"My dad is 85 years-old and he rides a bike. He*

*always encourages me to stay physically active"*. By contrary, others had some concerns, as Giovanna (OG) reported: *"My parents did not want me to run, they told me you will be too much tired, you have to recover"* or Nicoletta (OG) explained: *"My husband recommended me not to exaggerate, because I could get injured like my colleagues did."* Nevertheless, oncological patients described that friends, as well as the medical staff, support their choice to begin a running program. Daniela (OG) remembered: *"When I decided to start a running program, a lot of my friends texted me an encouraging message to continue exercising"* and Tony (OG) recounted: *"My oncologist told me that I had to do this, that after my cancer I had to rebuild my life"*.

**Environmental and organizational factors.**   For both groups, running in the natural environment is an important supportive factor to continue the activity. *"Sometimes I go running by the Garda lake, with a wonderful landscape, so it is a very pleasant environment for exercising. I feel less fatigue because I am concentrated on what my eyes see around me"* said Antonella (OG), or *"We live in a beautiful place that gives us the possibility to stay in touch with the nature and I like a lot running in this area"* Federica (HC) remembered. Moreover, OG recognized the great impact of training with an organized team, which provided them with a running campus, a trainer to indicate and explain them the workouts they needed to do: *"Have someone who follows you, like an organization, this is very motivating for me"* (Giulia, OG).

## Theme 2: Barriers

The interviews revealed various aspects that could interfere with the running EX. The identified barriers were categorized into four sub-themes, including: personal, interpersonal, organizational and community-policy factors (Table 3).

**Individual factors.**   The personal barriers recognized as obstacles to running were different between the two groups. The only common aspect was lack of time dedicated to running, although the perspective regarding this potential barrier was different between OG and HC. For healthy subjects, lack of time emerged as the principal obstacle that interferes with running: *"Unfortunately I must give priority to the work and when I was preparing for my half marathon and needed to run for two hours, I could run only one hour and a half"* (Erika, HC). Also from cancer survivors' point of view, lack of time in EX could be a potential barrier, but most of them explained how cancer disease changed this opinion: *"In a typical day it is difficult to cut out some time for EX because you have to work, prepare the dinner for your family, stay with your son because these are the priorities. After my cancer, I said to myself that now I exist! Now I can find my space and my time for EX, I demand it!"* (Antonella, OG).

In OG, a general consensus confirmed that injuries and treatment-related side effects represent potential obstacles for running. In particular, injuries of other training partners were indicated as reasons to discontinue running, how Elisa (OG) and Nicoletta (OG) reported: *"When I had a knee injury, I was strongly tempted to stop running, to give up the group"* and *"When four out of eight colleagues were injured, I thought of interrupting my training session because I did not want to hurt myself"*. Concerns about cancer- and treatment-related side effects were indicated as strong factors that may obstacle running: *"Hormonal therapy causes fatigue and joint pain, therefore sometimes it is very difficult for me to begin any exercise"* (Nadia, OG). Mirella (OG) also reported: *"My chemotherapy cycles were very long and hard. The main side effect that I experienced was peripheral neuropathy. Sometimes I had to interrupt running, because I had serious sensibility problem in my foots and I was afraid of hurting myself"*. Finally, HC reported that failing in pre-established running performance was a serious obstacle to maintain own training: *"When you expect to run for example 10 kilometres with a faster pace and you cannot do it, you lose confidence in yourself and sometimes the temptation to give up is really strong"* (Erika, HC).

**Interpersonal factors.** The OG referred that their trainers were not well prepared nor specifically qualified for advising a patient with oncological disease and this was a major obstacle. "*When I began to run my coach proposed me an overestimated program for my situation. After a month and a half my knees were blocked, I was in pain, I had difficulty to walk, I had to stop for one month and the temptation to interrupt was very strong*" (Antonella, OG). Another participant in the OG expressed concerns regarding the knowledge of some instructors: "*I did not have a good trainer, I never performed a warm-up phase, or exercised to reinforce my muscle, and also from a human point of view the support was completely missing*" (Ilaria, OG).

**Environmental and organizational factors.** Poor personal security and uncontrolled environment were interrelated and represented a barrier for running in both the HC and OG. "*I love running in the nature, but sometimes I meet weird people and I think: this way is not secure for running because I should run without listening to music in order to see if the person that stopped is following me*" recounted Lara during an interview in the healthy group. Also, Margherita (OG) told: "*I used to run on the bicycle lane and I always carried pepper spray with me because the environment was not controlled and I always had this feeling that someone was behind me, I did not feel comfortable*". However, this feeling of insecurity is magnified by poor maintenance of natural environment; in the OG: "*Some areas are poorly managed, there is tall grass that nobody cuts, the plants are not pruned and grow everywhere and consequently I'm afraid to run in those places*" (Rossella, OG). In addition, another problem for OG was air pollution: "*Sometimes I decide to postpone my training due to poor air quality; I do not want to breathe toxic air.*" (Ilaria OG). Another woman reported the difficulty to run in some areas because of air pollution: "*In some places, smog is very high and I have to admit that it is really difficult to go out for a run.*" (Margherita, OG).

**Community-policy factors.** Even if both groups recognized that the sport bodies organise several running manifestations, they agreed on the fact that the actual Italian policy situation was not favourable on promoting running. As Paola (HC) said: "*We live in a country where the main sport is football, the others are considered second class sports and, for this reason, are penalized*". Furthermore, the OG highlighted how the current traditionalist culture hindered the practice of PA in general: "*We live in a traditionalist culture, in which we teach our sons to go to school, to work, to have a family. These are the priorities.*" (Antonella, OG). Moreover, marketing was reported as a negative factor that blocks the correct and healthy promotion of running in OG. In fact, it usually appears that running EX is only adequate for athletes or for physically active subjects, and it is always related to body image. In this regard, Rossella (OG) and Nadia (OG) remembered: "*The current advertising and culture teach you to follow a woman model: lean, made up, that does not sweat; this is very disheartening for me.*" or "*Many information is incorrect and confounding; according to certain advertising you should train yourself to be cool and to have a beautiful body, not for health or for preventing or controlling chronic conditions.*"

## Discussion

To the best of our knowledge, this research represents the first qualitative investigation exploring motivations and barriers about running, as exercise training, in a group of female cancer survivors and compared them with matched healthy controls. We found several factors that stimulate the approach and adherence to running and others that limit them.

Regarding running motivations, several points were common in both groups, such as enjoyment, possibility to perform this type of EX in a natural environment, social support given by teammates and attitude towards EX. These results are in line with previous data [21]. *McIntosh et al.* for example identified physical and psychological benefits together with social support as factors that stimulated patients who have had cancer to maintain their walking

**Fig 1. Strategies to increase adherence and compliance in a running program.**

activity [18]. Nevertheless, from cancer survivors' perspective, other strong running motivations, related to their health history, were identified. Running performance was a challenge connected with their disease and a sort of demonstration they could overcome cancer, giving also hope to other cancer patients. Moreover, the focus group highlighted that patients who have had an oncological disease obtained more support from their family, friends, physician and workout teammates compared to healthy controls. This result is supported by *Husebø et al.*, who identified social support as a crucial component in influencing physical EX in women affected by breast cancer [22]. Regarding the environmental and organizational level, other motivations stimulated patients to maintain their running program, such as taking part in an organized training program and performing this activity in a natural environment. Doing EX outside is a common preference found in several other studies, in different cancer populations, while *Blaney et al.* reported that participating in an EX program, organized and supervised by an EX specialist was a strong motivator that seemed to offer assurance to survivors [23]. These findings support a series of recommendations that should be provided to cancer survivors in order to propose a successful running program, e.g. increase knowledge regarding EX benefits and promote group training, as summarized in Fig 1.

Focusing on barriers toward running, some environmental and organizational factors were similar between the oncological group and healthy subjects, such as poor personal security and untended environment. Another study has emphasized these obstacles mentioning that "safety

issues" were an impediment to patients affected by cancer walking activity [24]. In addition, they expressed many barriers related to their cancer journey [19, 23]. For example, cancer-related treatment side effects, such as fatigue, joint pain or peripheral neuropathy were identified as serious impediments significantly interfering with the maintenance of running EX. Moreover, physical injuries, inexperienced trainer, air pollution and the public scarce attractivity of running training have emerged as issues that can inhibit the adherence to a running program. Regarding EX security, a recent systematic review with metanalysis has investigated the safety and feasibility of EX among women affected by stage II-IV breast cancer. A total of 60 randomized controlled trials involving 5200 participants were included. The analysis showed no differences in adverse advents between EX and usual care, independently of EX supervision (EX supervised defined as over half of the Ex session involved face-to-face supervision) [25]. These findings support the EX safety, also in an unsupervised context, and therefore suggest that the fear of injuries observed in our oncological patients does not represent a real risk. Nevertheless, the psychological disease-related background might justify this concern. Indeed, a cancer diagnosis and its related treatments carry several physical and psychological impairments that alter the subject's perspectives, e.g. changes in body composition and body image, physical deconditioning. Cancer survivors might not feel confident or capable of performing EX, and specifically running, consequently, they are afraid to undergo injuries and want, for this reason, assurance regarding the trainer' professionality [26]. Therefore, the trainer should be able to reassure the participants about EX safety, personalizing the information and the instructions to provide. Moreover, after diagnosis, they usually search for additional information about their lifestyle (e.g. nutrition, smoking, alcohol consumption, PA) from several sources [27, 28]. Without adequate competence to correctly evaluate the quality of the collected information, there is the concrete risk of finding misleading news leading to unsafe and risky habits or that can induce excessive attention to those environmental factors potentially harmful as air pollution.

One last element seems significant, even if ambivalent. The possibility of reliving the positive emotions experienced in previous training experiences are indicated as significant motivations by the OG. This element further supports the promotion of exercise and training experiences also in the general population because its lack, may decrease the possibility of reaction in case of illness. Even in this case some suggestions, based on the identified barriers, should be considered while planning a running program for cancer survivors (Fig 1). Nowadays, some studies were conducted to improve EX adherence in cancer setting. Among them, Rogers and colleagues have proposed the BEAT trial (Better Exercise Adherence after Treatment) which aims to implement behaviour changes in breast cancer survivors by using the social cognitive theory. This dynamic model combines behavior, personal and environmental influences and, at the same time, includes barriers and facilitators in order to create a framework for the design of a durable physical activity intervention. In this study the participants were significantly more likely to meet physical activity recommendations both immediately post-intervention and after 3 months compared to control group, besides to show better improvements in fitness and quality of life [29]. These results confirm the importance of including EX barriers and motivators in planning an effective EX program. Focusing on running, some projects (i.e. "*Cancer to 5K*") proposed an EX training for cancer survivors, but not specific information regarding how the program was planned are available. To the best of our knowledge, some experiences have investigated the physical benefit of running in cancer[30], but no specific studies have organized the running program considering barriers and motivations.

Our study has some limitations as the low response rate especially in the healthy group. Although we cannot guarantee that the saturation principle was achieved in HC, our study

mainly focused on oncological patients' experiences and further investigations will be performed in order to confirm our findings. Moreover, it has to be acknowledged that the participants with cancer were already motivated to run as demonstrated by their participation in the R4S event. The oncological group was affected by different cancer types and considering the peculiarity of the physical EX evaluated (endurance running), the results are not widely generalizable to other activities. Nonetheless, precisely because these conditions represent a real-world situation, we believe that it is interesting to understand factors that induced these subjects to approach and adhere to running EX.

In conclusion, the current literature shows the strong importance of a constant PA, such as endurance running, after a cancer diagnosis in order to reduce recurrence risk and mortality. Exploring the factors that limit and favour the promotion of an active lifestyle is extremely important to design specific interventions. Our study investigated, using an ecological approach, barriers and motivations towards endurance running in women affected by cancer and compared them with matched healthy subjects. We found that OG had many motivations originating by personal and interpersonal levels. Furthermore, they interfaced with several obstacles, present into all four levels of the ecological model. Among them, the cancer experience appeared significantly important and influenced both motivators and barriers. Developing a running program that considers all these aspects, may increase its success in terms of both adherence and compliance in this kind of patients (Fig 1).

## Acknowledgments

We thank all the participants that took place in this study.

## Author Contributions

**Conceptualization:** Alice Avancini, Massimo Lanza.

**Data curation:** Alice Avancini, Daniela Tregnago, Paolo Frada.

**Formal analysis:** Alice Avancini, Kristina Skroce, Paolo Frada.

**Funding acquisition:** Massimo Lanza.

**Investigation:** Alice Avancini, Kristina Skroce.

**Methodology:** Alice Avancini, Daniela Tregnago, Sara Pilotto, Massimo Lanza.

**Project administration:** Cantor Tarperi, Federico Schena, Michele Milella, Massimo Lanza.

**Resources:** Alice Avancini, Kristina Skroce, Daniela Tregnago, Ilaria Trestini, Clelia Bonaiuto, Cantor Tarperi, Federico Schena, Sara Pilotto.

**Software:** Alice Avancini, Paolo Frada.

**Supervision:** Federico Schena, Michele Milella, Massimo Lanza.

**Validation:** Massimo Lanza.

**Visualization:** Alice Avancini, Kristina Skroce, Daniela Tregnago, Ilaria Trestini, Maria Cecilia Cercato, Clelia Bonaiuto, Federico Schena, Michele Milella, Sara Pilotto, Massimo Lanza.

**Writing – review & editing:** Daniela Tregnago.

**Writing – original draft:** Alice Avancini, Kristina Skroce, Sara Pilotto.

**Writing – review & editing:** Alice Avancini, Kristina Skroce, Paolo Frada, Ilaria Trestini, Maria Cecilia Cercato, Clelia Bonaiuto, Cantor Tarperi, Federico Schena, Michele Milella, Sara Pilotto, Massimo Lanza.

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
