## [Decision Letter · Decision Letter 0]

3 Feb 2020

PONE-D-19-35627

“Running with cancer”: a qualitative study to evaluate barriers and motivations in running for female oncological patients.

PLOS ONE

Dear Dr Avancini,

Thank you for submitting your manuscript to PLOS ONE. After careful consideration, we feel that it has merit but does not fully meet PLOS ONE’s publication criteria as it currently stands. Therefore, we invite you to submit a revised version of the manuscript that addresses the points raised during the review process.

This is a straightforward study to start to gain insight into the opinions and experiences of people with cancer who are motivated to exercise through running.

The manuscript should state how many focus groups were held, how many participants were in each: it looks like there were two focus groups but this should be stated clearly.

Avoid using the terms “experimental group” and “control group” that are more familiar in a quantitative context.

In the discussion of limitations the authors should consider the issue that the participants with cancer were already motivated to run as seen by their participation in the Run for Science event.

Data saturation: I’m not convinced by the implication that data saturation was shown to be reached. Yes, it is possible that there were no further insights emerging from each focus group but there is a real possibility that further groups would have. The authors should revisit their thoughts on this and discuss the limitations of having only one group for each group.

We would appreciate receiving your revised manuscript by 1 April 2020. To enhance the reproducibility of your results, we recommend that if applicable you deposit your laboratory protocols in protocols.io, where a protocol can be assigned its own identifier (DOI) such that it can be cited independently in the future. For instructions see: http://journals.plos.org/plosone/s/submission-guidelines#loc-laboratory-protocols

We look forward to receiving your revised manuscript.

Kind regards,

Denis Martin, PhD

Academic Editor

PLOS ONE

Additional Editor Comments (if provided):

This is a straightforward and well-written study to start to gain insight into the opinions and experiences of people with cancer who are motivated to exercise through running. The reviewer has provided very useful points and each of these should be addressed. In addition, there are some other points below that should also be addressed.

The manuscript should state how many focus groups were held, how many participants were in each: it looks like there were two focus groups but this should be stated clearly.

Avoid using the terms “experimental group” and “control group” that are more familiar in a quantitative context.

In the discussion of limitations the authors should consider the issue that the participants with cancer were already motivated to run as seen by their participation in the Run for Science event.

Data saturation: I’m not convinced by the implication that data saturation was shown to be reached. Yes, it is possible that there were no further insights emerging from each focus group but there is a real possibility that further groups would have. The authors should revisit their thoughts on this and discuss the limitations of having only one group for each group.

Journal Requirements:

"The authors received no specific funding for this work."

Reviewers' comments:

Reviewer's Responses to Questions

**Comments to the Author**

1. Is the manuscript technically sound, and do the data support the conclusions?

Reviewer #1: Yes

2. Has the statistical analysis been performed appropriately and rigorously? 

Reviewer #1: Yes

3. Have the authors made all data underlying the findings in their manuscript fully available?

Reviewer #1: Yes

4. Is the manuscript presented in an intelligible fashion and written in standard English?

Reviewer #1: Yes

5. Review Comments to the Author

Reviewer #1: Dear Alice and co-authors,

Thank you for the opportunity to review your qualitative paper. This was a very interesting paper exploring the perceived motivations and barriers for running/exercise by women with cancer. Below I have outlined some suggestions which I feel will improve the quality of the paper.

Major revision suggestions:

1. It appears that the use of the social ecological model was employed during the set-up of the project (i.e., used to frame the interview questions and the way the results were analyzed/categorized). Personally, I do not have an issue with this, I believe you have analyzed and summarized the issues in a logical way that can be easily understood by many. However, some qualitative researchers may argue that themes did not organically emerge as the model was utilized to both structure the interviews and the analysis. Therefore, I would suggest refraining from saying that themes emerged or were ‘found’ – rather, you analyzed the transcripts with this model in mind and categorized the common themes to reflect the levels within the model.

2. Methods: How was informed consent obtained? Written or verbal? Please include a more specific (brief) explanation in this section.

3. Discussion: I would have liked to see a bit more discussion on the issue of safety for performing exercise as a cancer patient. Obviously, exercise is recommended, however for some survivors/patients running may not be safe without an experienced supervisor/trainer. This theme did come up in your results so it would be great to further discuss the safety issues regarding exercise in cancer patients and perhaps some suggestions/recommendations for interventions.

4. Figure 1: I really like that you have included this to explain the recommendations based on your findings, however I’d love for this to be mentioned earlier in the discussion section as well as explained in more depth (see point 3 above). For instance, do any running programs for cancer survivors exist (both evident in peer review literature and outside of this), that employ some of these suggestions?

5. Some minor English/grammar issues throughout which need to be addressed; see below for further specific suggestions.

Minor revision suggestions:

Abstract

• Lines 31-34: Adjust description of themes to reflect suggestion above (i.e., themes were categorized into the social ecological model, rather than emerged as such).

• Line 37/38: Change ‘Running program’ to ‘Running programs’

• Lines 36-40: Last sentence is very long, I’d considered revising into 2.

Introduction

• Line 45: is this number correct? Do you mean 1,837,412? Further, is this the number of cancer patients/survivors in Italy? I’d revise this sentence.

• Line 62: include ‘the’ between also and cancer setting

• Line 74: please fix reference

• Line 78: please rewrite this sentence as the part ‘that approaching running’ doesn’t make sense.

Method

• Line 98: please change ‘and aged major than 18 aged’ to ‘and 18 years of age or older’

• Line 121: please change ‘the first and last author’ to the appropriate initials.

• Line 129: please change the word ‘proposed’ to ‘provided to participants to complete’.

• Line 140-142: The description of triangulation could be refined. In my understanding, triangulation of data reflects the comparison of differing sources (e.g., control Vs experimental group) as a form of qualitative validity. In this context, I would expect that you have compared themes from the control & oncology group to find similarities & differences. I would suggest stating that this here.

Results

• Line 145: please provide the number of participants in the cancer group.

• 147: I would suggest changing the word ‘identified’ to ‘were categorized into’, as the interview questions specifically focused on motivations/barriers (i.e., you did not identify this as a theme during analysis, rather you focused the discussion on these issues).

• Table 2: it’s unclear to me how to interpret the ‘Family Income’ levels. I would suggest providing more information on this scale in the methods section.

• Table 2: You have written ‘undergoing’ treatment for 2 of the sections however no participants were currently undergoing treatment (as seen in the ‘current treatment status’ section). I would remove these words in the hormone therapy & others treatment box.

• Table 3: Could you change the category ‘friends’ (under barriers for controls) to lack of social support?

• Table 3: Could you change the category ‘run as second class sport’ (under barriers for controls) to something more specific? It’s a bit unclear from the table what this means (e.g., ‘running perceived negatively by others’)

• 164: Please include ‘such as’ before the word ‘enjoyment’

• Line 176: please change the word ‘differently’ to ‘In contrast’

• line 198-199: please revise this sentence, the grammar is incorrect

• line 203: please change the word ‘referred’ to ‘explained’ or similar

• line 263: please change the word ‘consent’ to ‘consensus’

• line 243: please use the full word ‘exercise’ as this is a direct quote.

• Line 284: please revise this sentence as the grammar is incorrect.

Discussion/Conclusion

• Line 301: please remove ‘a’ from ‘have had a cancer’

• Line 327-329: could you please provide a reference for this.

• Lines 333-335: please revise this sentence as the English/grammar is not quite correct.

6. PLOS authors have the option to publish the peer review history of their article (what does this mean?). If published, this will include your full peer review and any attached files.

Reviewer #1: No

---

## [Author Response · Author response to Decision Letter 0]

11 Feb 2020

Verona, February 9th, 2020

Dear Prof. Denis Martin,

 Enclosed please find a thoroughly revised version of the manuscript entitled “RUNNING WITH CANCER”: A QUALITATIVE STUDY TO EVALUATE BARRIERS AND MOTIVATIONS IN RUNNING FOR FEMALE ONCOLOGICAL PATIENTS.” [PONE-D-19-35627] that we wish to resubmit for publication in Plos One. 

We would like to thank the reviewers for the thoughtful and stimulating comments that have prompted us to update and clarify several points and revise the manuscript accordingly. 

We thus hope that the quality of the manuscript has now substantially improved, so that it may be reconsidered for publication.

A point-by-point rebuttal description of the performed revisions follows herein:

Editor: all the revisions suggested by the editor has been performed.

 Editor’ Comments: Response:

1) The manuscript should state how many focus groups were held, how many participants were in each: it looks like there were two focus groups but this should be stated clearly. Three and two focus groups for oncological and healthy subjects were performed, respectively. A detailed description was added in “data collection”, lines 116/117.

2) Avoid using the terms “experimental group” and “control group” that are more familiar in a quantitative context. As correctly suggested by the editor, we have changed the names of the groups in ‘oncological group’ and ‘healthy subjects’. 

3) In the discussion of limitations the authors should consider the issue that the participants with cancer were already motivated to run as seen by their participation in the Run for Science event. Done, lines 382-384.

4) Data saturation: I’m not convinced by the implication that data saturation was shown to be reached. Yes, it is possible that there were no further insights emerging from each focus group but there is a real possibility that further groups would have. The authors should revisit their thoughts on this and discuss the limitations of having only one group for each group. We have modified the group number as reported in point 1 and revisited the limitations according to your suggestion (lines 380-382)

Referee #1: all the revisions suggested by the reviewer have been performed. 

 Reviewer’ Comments: Response:

1) It appears that the use of the social ecological model was employed during the set-up of the project (i.e., used to frame the interview questions and the way the results were analyzed/ categorized). Personally, I do not have an issue with this, I believe you have analyzed and summarized the issues in a logical way that can be easily understood by many. However, some qualitative researchers may argue that themes did not organically emerge as the model was utilized to both structure the interviews and the analysis. Therefore, I would suggest refraining from saying that themes emerged or were ‘found’ – rather, you analyzed the transcripts with this model in mind and categorized the common themes to reflect the levels within the model. Thanks for your precious suggestion. We have modified accordingly. 

2) How was informed consent obtained? Written or verbal? Please include a more specific (brief) explanation in this section. Done

3) I would have liked to see a bit more discussion on the issue of safety for performing exercise as a cancer patient. Obviously, exercise is recommended, however for some survivors/ patients running may not be safe without an experienced supervisor/trainer. This theme did come up in your results so it would be great to further discuss the safety issues regarding exercise in cancer patients and perhaps some suggestions/recommendations for interventions. Thanks for your suggestion. We have implemented the discussion with the recommended theme. (lines 341-348)

4) Figure 1: I really like that you have included this to explain the recommendations based on your findings, however I’d love for this to be mentioned earlier in the discussion section as well as explained in more depth (see point 3 above). For instance, do any running programs for cancer survivors exist (both evident in peer review literature and outside of this), that employ some of these suggestions? Thanks again for your suggestion. We have implemented the discussion about. (lines 329-332; 353-355; 364-378)

5) Lines 31-34: Adjust description of themes to reflect suggestion above (i.e., themes were categorized into the social ecological model, rather than emerged as such). Done

6) Line 37/38: Change ‘Running program’ to ‘Running programs’ Done

7) Lines 36-40: Last sentence is very long, I’d considered revising into 2. Done

8) Line 45: is this number correct? Do you mean 1,837,412? Further, is this the number of cancer patients/survivors in Italy? I’d revise this sentence. Done

9) Line 62: include ‘the’ between also and cancer setting Done

10) Line 74: please fix reference Done

11) Line 78: please rewrite this sentence as the part ‘that approaching running’ doesn’t make sense. Done

12) Line 98: please change ‘and aged major than 18 aged’ to ‘and 18 years of age or older’ Done

13) Line 121: please change ‘the first and last author’ to the appropriate initials. Done

14) Line 129: please change the word ‘proposed’ to ‘provided to participants to complete’. Done

15) Line 140-142: The description of triangulation could be refined. In my understanding, triangulation of data reflects the comparison of differing sources (e.g., control Vs experimental group) as a form of qualitative validity. In this context, I would expect that you have compared themes from the control & oncology group to find suggest stating that this here. similarities & differences. Done

16) Line 145: please provide the number of participants in the cancer group. Done

17) Line 147: I would suggest changing the word ‘identified’ to ‘were categorized into’, as the interview questions specifically focused on motivations/barriers (i.e., you did not identify this as a theme during analysis, rather you focused the discussion on these issues). Done

18) Table 2: it’s unclear to me how to interpret the ‘Family Income’ levels. I would suggest providing more information on this scale in the methods section. Done

19) Table 2: You have written ‘undergoing’ treatment for 2 of the sections however no participants were currently undergoing treatment (as seen in the ‘current treatment status’ section). I would remove these words in the hormone therapy & others treatment box. Done

20) Table 3: Could you change the category ‘friends’ (under barriers for controls) to lack of social support? Done

21) Table 3: Could you change the category ‘run as second class sport’ (under barriers for controls) to something more specific? It’s a bit unclear from the table what this means (e.g., ‘running perceived negatively by others’) Done

22) Line 164: Please include ‘such as’ before the word ‘enjoyment’ Done

23) Line 176: please change the word ‘differently’ to ‘In contrast’ Done

24) line 198-199: please revise this sentence, the grammar is incorrect Done

25) line 203: please change the word ‘referred’ to ‘explained’ or similar Done

26) line 263: please change the word ‘consent’ to ‘consensus’ Done

27) line 243: please use the full word ‘exercise’ as this is a direct quote. Done

28) Line 284: please revise this sentence as the grammar is incorrect. Done

29) Line 301: please remove ‘a’ from ‘have had a cancer’ Done

30) Line 327-329: could you please provide a reference for this. Done

31) Lines 333-335: please revise this sentence as the English/grammar is not quite correct Done

Moreover, we have checked carefully the author checklist in order to provide all the requested manuscript details.

We look forward to hearing from you regarding our submission. We would be glad to respond to any further questions and comments that you may have. 

Best regards,

Alice Avancini, Sara Pilotto, and Massimo Lanza on beha

---

## [Decision Letter · Decision Letter 1]

13 Mar 2020

“Running with cancer”: a qualitative study to evaluate barriers and motivations in running for female oncological patients.

PONE-D-19-35627R1

Dear Dr. Avancini,

We are pleased to inform you that your manuscript has been judged scientifically suitable for publication and will be formally accepted for publication once it complies with all outstanding technical requirements.

With kind regards,

Denis Martin, PhD

Academic Editor

PLOS ONE

Additional Editor Comments (optional):

Reviewers' comments:

Reviewer's Responses to Questions

**Comments to the Author**

1. If the authors have adequately addressed your comments raised in a previous round of review and you feel that this manuscript is now acceptable for publication, you may indicate that here to bypass the “Comments to the Author” section, enter your conflict of interest statement in the “Confidential to Editor” section, and submit your "Accept" recommendation.

Reviewer #1: All comments have been addressed

2. Is the manuscript technically sound, and do the data support the conclusions?

Reviewer #1: Yes

3. Has the statistical analysis been performed appropriately and rigorously? 

Reviewer #1: Yes

4. Have the authors made all data underlying the findings in their manuscript fully available?

Reviewer #1: (No Response)

5. Is the manuscript presented in an intelligible fashion and written in standard English?

Reviewer #1: Yes

6. Review Comments to the Author

Reviewer #1: Thank you for addressing the review comments, I enjoyed reading the revised version and am happy to recommend the paper for publication. Just a few minor grammatical errors which I would recommend attending to before publication;

1. In Table 2, I would change the comma (,) to a full-stop (0.2) for decimal places (unless this is standard for the journal).

2. Lines 190-191: you have used two colons in the same sentence. I'd revise.

3. Line 349-350: I would replace the words 'fake news' and 'wrong' to 'misleading and 'unsafe', respectively (the current wording is too casual in my opinion).

7. PLOS authors have the option to publish the peer review history of their article (what does this mean?). If published, this will include your full peer review and any attached files.

Reviewer #1: No

---

## [Editor Report · Acceptance letter]

20 Mar 2020

PONE-D-19-35627R1 

“Running with cancer”: a qualitative study to evaluate barriers and motivations in running for female oncological patients. 

Dear Dr. Avancini:

I am pleased to inform you that your manuscript has been deemed suitable for publication in PLOS ONE. Congratulations! Your manuscript is now with our production department. 

With kind regards,

on behalf of

Dr. Denis Martin 

Academic Editor

PLOS ONE